# Organic light emitting board for dynamic interactive display

Eui Hyuk Kim[1], Sung Hwan Cho[1], Ju Han Lee[1], Beomjin Jeong[1], Richard Hahnkee Kim[1], Seunggun Yu[1], Tae-Woo Lee[2], Wooyoung Shim[1] & Cheolmin Park[1]

Interactive displays involve the interfacing of a stimuli-responsive sensor with a visual human-readable response. Here, we describe a polymeric electroluminescence-based stimuli-responsive display method that simultaneously detects external stimuli and visualizes the stimulant object. This organic light-emitting board is capable of both sensing and direct visualization of a variety of conductive information. Simultaneous sensing and visualization of the conductive substance is achieved when the conductive object is coupled with the light emissive material layer on application of alternating current. A variety of conductive materials can be detected regardless of their work functions, and thus information written by a conductive pen is clearly visualized, as is a human fingerprint with natural conductivity. Furthermore, we demonstrate that integration of the organic light-emitting board with a fluidic channel readily allows for dynamic monitoring of metallic liquid flow through the channel, which may be suitable for biological detection and imaging applications.

[1] Department of Materials Science and Engineering, Yonsei University, Seoul 120-749, Korea. [2] Department of Materials Science and Engineering, Seoul National University, 1 Gwanak-ro, Gwanak-gu, Seoul 08826, Korea. Correspondence and requests for materials should be addressed to C.P. (email: cmpark@yonsei.ac.kr).

Accurate and real-time display capabilities are important for the realization of interactive displays[1-7], which generally incorporate active sensor arrays to achieve efficient stimuli-interfacing, as well as optical elements for visualizing the stimuli. Adaptive or interactive displays coupled with stimuli-responsive sensors have been demonstrated using pneumatic microfluidic networks[1,2], organic electrochromics[3,4] and thermochromics[5-7]. These efforts provided results supporting the importance of display integration with sensors, although the optical display elements are based on absorbance and reflective modes which in general suffer from rather low brightness, slow response time and low light efficiency. In this context, organic light-emitting diodes (OLEDs) are promising because they are ultra-thin and have high-colour contrast and efficiency[8-15]. Hybrid thin-film transistor and pressure sensor array structures with pixelated OLEDs that can quantitatively correlate the applied pressure to OLED arrays have been demonstrated[16]. This approach still requires efforts for ideal interactive displays that could (i) readily adopt the dynamic stimuli with high temporal and spatial resolution and (ii) be simply fabricated without an active-matrix. A few studies have proceeded beyond complex-circuit demonstrations[17-19] and towards functionality and simplicity of fabrication[20-22] (Supplementary Table. 1).

Several important features should be further considered for a dynamic interactive display. First, a device would be advantageous if it responds instantaneously to other stimuli, that is, conductance rather than pressure and directly visualizes the spatial information of the conductance. This requires a device with its architecture where the interaction between the light-emitting component and stimulus is directly interactive in the sense of (i) light emission and (ii) stimuli sensing. Second, the state of this interaction should be maintained as long as the stimulus remains to ensure accurate information display. Third, the device structure and fabrication should be simplified as far as possible. To this end, we envisioned that a field-driven organic light-emitting platform would be useful for dynamic interactive display applications. One or two insulating layers can be inserted between the electrode and light-emitting layer, to facilitate carrier injection on alternating-current field[23-36]. In addition, this unique architecture with insulating layers does not permit the direct contact between the electrodes and light-emitting layers and thus band alignment design of the two layers may not be necessarily considered. Therefore, this device architecture with a diverse material of choice could lead to the sensing and display of a variety of conductive information, through electroluminescence (EL) by an alternating-current electric field.

Here we report a powerful approach for achieving a dynamic interactive display to directly visualize conductive information based on an alternating-current electroluminescent (AC EL) device. Our field-induced devices provide potential advantages in terms of simplicity, functionality, and broad applicability compared to previous OLED-based interactive displays that involved active elements. Our organic light-emitting board (OLEB) consists of stacked layers of $SiO_2$, Poly(3,4-ethylenedioxythiophene)-poly(styrenesulfonate) (PEDOT:PSS), a light-emitting layer, polyethylenimine (PEI) and Zinc oxide (ZnO) on two transparent indium tin oxide (ITO) electrodes separated by an in-plane gap. The light emission properties depend on the conductivities of top conductive materials up to certain conductivity. The OLEB is capable of sensing and directly visualizing conductive information in the form of symbols, characters, arbitrary patterns and even the dynamic flow of a conductive substance, potentially making it suitable as an *in-situ* sensing board as schematically shown in Fig. 1a. The OLEB can be fabricated on a flexible substrate, enabling conformal wrapping, for example, to human fingers, such that fingerprint detection and imaging can be achieved owing to skin conductance. Below we describe the working mechanism and performance of the OLEB devices and then demonstrate potential applications for dynamic interactive displays.

## Results

**Operation principle and light emission performance.** The structure of our field-induced AC EL platform is shown in Fig. 1b. Light is not emitted in a device on alternating-current field between two parallel ITO electrodes without a top electrode on the $SiO_2$ layer because the in-plane electric field is only developed between two edges of the electrodes, and an electric field is rarely exerted on an emitting layer placed on top of the electrodes. Interestingly, when an electrode is placed on top of the $SiO_2$ layer across the two bottom electrodes, the device emits light in the two cross-overlapped areas. Efficient light emission with a non-contact top floating electrode in our AC EL platform will be examined in detail later, and the principle is also applicable when a finger with natural conductivity is in contact as a floating electrode on the $SiO_2$ surface, giving rise to direct visualization of fingerprints. The environmentally sensitive polymeric light-emitting layer should be appropriately protected not only from air but also from undesirable contaminants; thus, it was encapsulated with a transparent and chemically robust $SiO_2$ insulating layer.

In our device, the carriers are directly injected from the bottom ITO electrodes. To optimize light emission, we adopted a device architecture reminiscent of an inverted OLED with stacked layers of an electron injection layer of ZnO/PEI (10 nm in thickness), an emitting layer of Super Yellow (PDY-132)/Multi-walled carbon nanotubes (MWNTs) composite (90 nm), a hole injection layer of PEDOT:PSS (60 nm), an $SiO_2$ insulating layer (300 nm), and an Al top electrode (75 nm) on two in-plane ITO bottom electrodes (left, Fig. 1b). Successful fabrication was confirmed by visualizing the device cross-section produced by a focused ion beam (right, Fig. 1b). Energy dispersive X-ray mapping of the constituent atomic elements of the layers clearly shows that individual layers of our device were well developed with sharp interfaces (Supplementary Fig. 1). The surface morphology of individual layers was also analysed by atomic force microscopy (AFM); the results show that all of the layers exhibit low root-mean-squared (RMS) roughness (below 5 nm, Supplementary Fig. 2) indicating that no damage occurred during the successive spin-coating of layers. ZnO/PEI was used as electron injection layers from ITO to Super Yellow/MWNT as emitting layer. A PEI layer was introduced to reduce the injection barrier between ZnO and emitting layer because of its strong dipole moment[37-40]. The MWNTs were employed as an emitting layer to improve light emission by lowering injection barriers for both holes and electrons[24,25].

When an alternating-current field is applied between the two in-plane ITO electrodes, light emission of the two light-emitting units (LEUs) overlapped with the floating Al electrode should occur sequentially and repeatedly. The results under alternating-current field demonstrate that the LEU in the left hand side was turned on with a negative polarity electric field on the left ITO electrode while no light was emitted at the LEU in the right hand side with a positive polarity field, as shown in Fig. 1c. When the polarity was switched from negative to positive on the left ITO electrode (positive to negative on the right ITO one), the right LEU was turned on. Sequential alternating-current fields gave rise to alternating light emission in the two LEUs. The light emission of our device on in-plane alternating-current can be explained with a vertically stacked, equivalent model with a mirror plane at

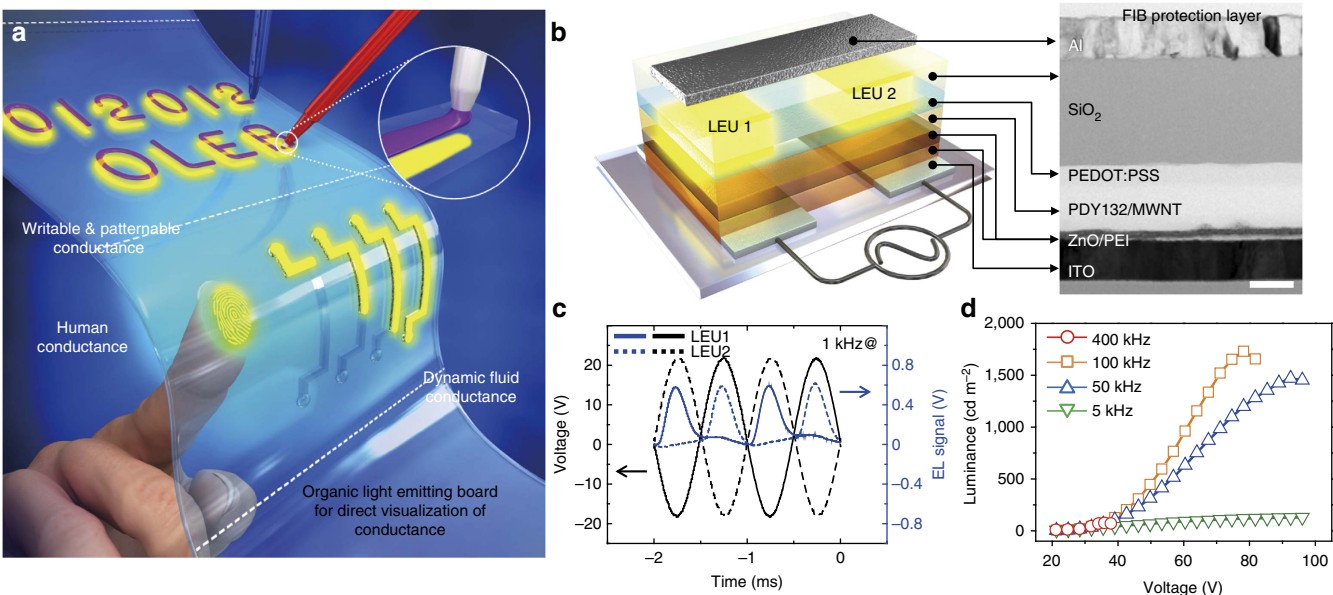

**Figure 1 | Device architecture and working principle and device performance of the organic light-emitting board (OLEB).** (**a**) Conceptual illustration of an OLEB suitable for direct visualization of conductance. (**b**) Schematic of a device structure of a parallel alternating-current electroluminescent (AC EL) and a high-resolution cross-sectional transmission electron microscopy (TEM) image of the device. The scale bar is 100 nm. (**c**) The time-resolved EL signals of two light-emitting units (LEUs) of a parallel AC EL device under alternating-current voltage with a frequency of 1 kHz. Solid and dot black lines correspond to the electric fields imposed to (LEU) 1 and 2, respectively on alternating-current operation. Light emits only at the negative polarity for each LEU. (**d**) Luminance versus voltage ($L$–$V$) characteristic of a parallel AC EL device at different frequencies.

the centre of a floating Al electrode with top and bottom ITO electrodes (Supplementary Fig. 3). The voltage bias developed between ITO and Al electrodes on alternating-current input applied at the two ITO electrodes was independently measured to confirm our speculation (Supplementary Fig. 4). As expected from the model, half of the input voltage bias at the two parallel electrodes was observed between the ITO and Al contacts at every alternating voltage input (Supplementary Fig. 4b). It is noted that the device architecture is very efficient compared with a vertical contact device, because both polarities of alternating-current field can be utilized, allowing for light emission of two LEUs on one alternating-current cycle. Both left and right units appear to remain turned on when the time delay between two light-emitting units ($\Delta t = t_1 - t_2$) is faster than the persistence of vision at frequencies greater than 100 Hz.

The light emission performance as a function of applied voltage at different alternating-current frequencies is shown in Fig. 1d. The characteristic EL of Super Yellow/MWNTs composite was observed with a maximum intensity at 585 nm (Supplementary Fig. 5). Similar to a conventional AC EL device with vertical contacts, the light intensity of each LEU increases with the voltage[24–27,32]. The light intensity also increases with alternating-current frequency at a constant voltage bias, and luminance of 2,000 cd m$^{-2}$ was achieved at 100 kHz. However, very poor light emission was observed at 400 kHz owing to insufficient time for carrier injection, consistent with previous results[31,32]. It should be noted that the device was electrically broken down at the voltage of ~60 V. The device performance was degraded at 200 kHz, compared with that at 100 kHz and the device failure occurred at ~60 V, when it was operated both 300 and 400 kHz (Supplementary Fig. 6). The current density increases with both applied voltage and frequency. The frequency-dependent light emission efficiency was also examined in detail (Supplementary Fig. 7). Two control experiments were additionally performed to examine the roles of not only MWNTs but also ion doping in ZnO layer (Supplementary Fig. 8). A parallel AC EL device without MWNTs showed the performance

much worse than one with nanotubes. In addition, a parallel AC EL device in which ZnO layer was doped with Li ions exhibited its performance worse than one without doping. As expected from the proposed mechanism, we confirmed that the device performance of a parallel AC EL device was rarely varied with the distance of two ITO bottom electrodes (Supplementary Fig. 9). The turn-on voltage was further reduced when ca. 100 nm thick SiO$_2$ layer was employed instead of 300 nm thick layer. The parallel AC EL device was turned on at ~10 V (Supplementary Fig. 10). The results suggest that our device can be operated at low voltage with further optimization. The measured light intensity and current of a thousand cd m$^{-2}$ and mA, respectively, are sufficiently high that our device is suitable as a OLEB capable of imaging conductive objects.

**Visualization of writable and patternable conductance.** The main advantage of our device is that the light emission is independent of the top electrode material because the hole injection is field-driven and facilitated by a PEDOT:PSS layer that is separated by an SiO$_2$ layer from the top electrode. Four different metal electrodes were employed including Ag, Au, Al and Cu, as shown in Fig. 2a. All four electrodes were equally shared with two parallel ITO bottom electrodes, allowing simultaneous comparison of the light emission performance. The luminance-voltage results of the devices with different electrodes were similar within experimental uncertainty, as shown in Fig. 2b, confirming the light emission principle of our devices (Supplementary Fig. 11). The luminance and current density values of the devices were similar at a constant voltage and frequency irrespective of the metal used, as shown in Fig. 2c. We observed that a highly conductive PEDOT:PSS film was also employed as a top electrode, giving rise to light emission. The device performance with the PEDOT:PSS electrode was, however, inferior to that with a conventional metal one due to the lower conductivity of the PEDOT:PSS layer (Supplementary Fig. 12)[41]. The detailed study on the maximum luminance as a function of conductivity of the materials on top of

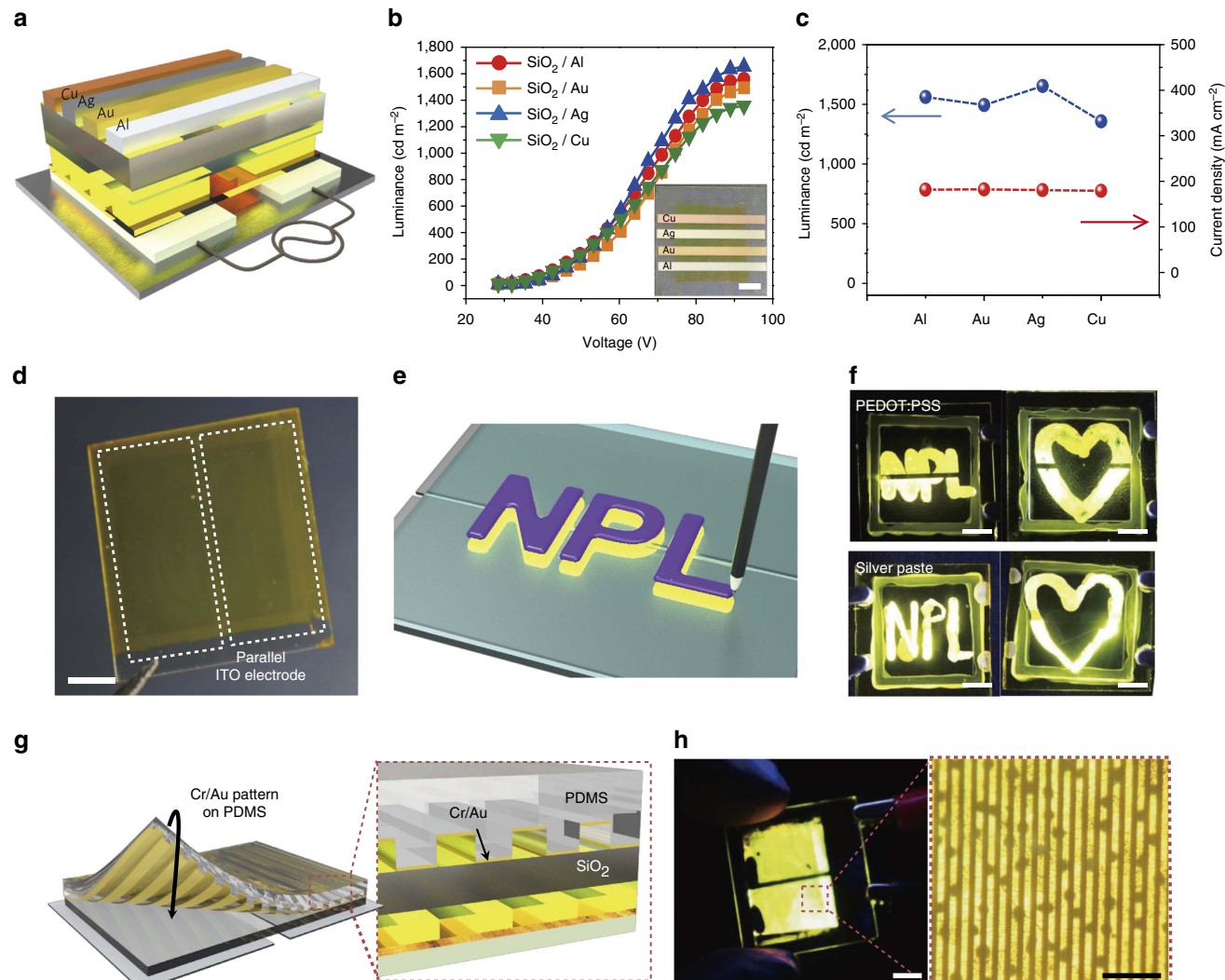

**Figure 2 | Direct visualization of writable and patternable conductance on OLEB.** (**a**) Schematic of parallel alternating-current electroluminescent (AC EL) devices with four different metal electrodes. (**b**) Luminance versus voltage characteristics (L–V) of the devices shown in a photograph in the inset. The scale bar is 5 mm. (**c**) Maximum luminance and current density characteristics of the devices with four metal electrodes. (**d**) Photograph of a field-induced organic light emitting board (OLEB) with two parallel indium tin oxide (ITO) bottom electrodes guided with dotted boxes in white. The scale bar is 5 mm. (**e**) Schematic of a writable OLEB using a conductive pencil. (**f**) Photographs of electroluminescent (EL) images visualizing metallic characters and symbols written by both PEDOT:PSS ink (top two photographs) and silver pastes (bottom two photographs). The scale bar is 5 mm. (**g**) Schematics of visualization of a metallic micropattern on the board by physical contact. (**h**) A photograph of an OLEB on which a patterned Au on a poly(dimethyl siloxane) (PDMS) substrate was in conformal contact. An optical microscope (OM) image on the right side shows the patterned EL arising from the metallic pattern on the board. The scale bar is 5 mm (left) and 50 μm (right). All the results were obtained with the alternating-current frequency of 100 kHz.

an insulating layer shows that the emission property was proportional to the conductivity of the materials up to $700\,S\,cm^{-1}$ (Supplementary Fig. 12). The results suggest that our device can also quantify the conductivity at the low conductance regime. For example, the same symbols written with PEDOT:PSS inks with different conductivities would exhibit different EL intensity although the spatial information (that is, registration of the symbols) is the same.

Moreover, the conductance related light emission offers a useful way for realizing a device on which any conductive information can be visualized by EL when written across the two parallel bottom electrodes, as indicated with two white dotted boxes in Fig. 2d. For example, once patterns can be written with conducting ink and the information visualized as per Fig. 2e. High-conductivity PEDOT:PSS and Ag paste inks were used to write symbols on the $SiO_2$ surface, and fluorescent light corresponding to the written conductive information was emitted, as shown in

Fig. 2f. It should be noted that small gaps in the patterns written by PEDOT:PSS ink (top, Fig. 2f) arose from the gap between the two parallel ITO electrodes. However, these gaps can be made invisible either by enhancing brightness of the patterns with an increase in voltage or by reducing the electrode gap. In fact, the gaps in fluorescent images drawn with Ag ink disappeared when a higher voltage was applied (bottom, Fig. 2f). To have more quantitative relation between the gap and the voltage required for gap disappearance, we fabricated several parallel AC EL devices with different gaps of 1 mm, 0.5 mm, 0.1 mm, 0.05 mm, 0.025 mm, 0.01 mm by conventional photolithography, followed by metal deposition and lift-off (Supplementary Fig. 13). A gap became invisible with its width of 50 μm and below when the device was operated at the constant voltage of 30 V, which was close to the turn-on voltage of the device. The results clearly show that the gap in our parallel AC EL device is not a critical issue but a minor drawback for further development.

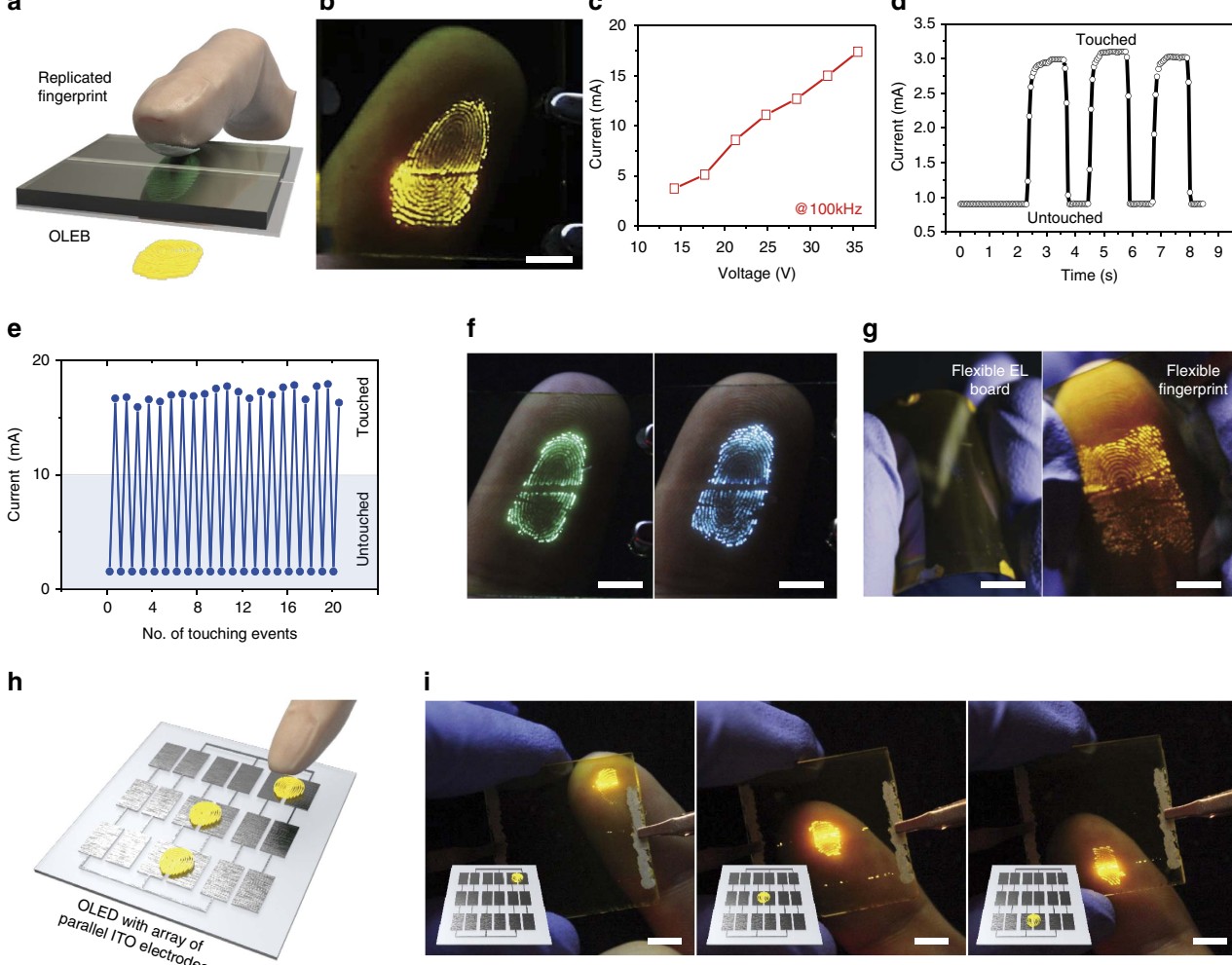

**Figure 3 | Simultaneous detection and imaging of fingerprint.** (**a**) Schematic of a field-induced organic light emitting board (OLEB) based on a parallel alternating-current electroluminescent (AC EL) device for visualizing fingerprint. (**b**) A photograph of an electroluminescent (EL) image of fingerprint pattern obtained by gentle contact of a finger on an OLEB. The scale bar is 5 mm. (**c**) A plot of device current as a function of the applied voltage on finger contact at 100 kHz. (**d**) The variation of device current under repeated events of finger touching and de-touching. (**e**) Reliable variation of the device current as a function of the number of finger touching events on a SiO₂ insulating layer of the OLEB. (**f**) Photographs of EL images of fingerprint patterns obtained from OLEBs with green and blue fluorescent polymers under gentle finger contact. The scale bar is 5 mm. (**g**) Photographs of a mechanically flexible OLEB and a fingerprint EL image obtained from the flexible EL board. The scale bar is 5 mm. (**h**) Schematic of an OLEB consisting of 3 × 3 arrays of parallel-type Indium tin oxide (ITO) electrodes. (**i**) Photographs of an OLEB with arrays of parallel-type ITO electrodes with position recognition of a finger touch. The scale bar is 5 mm.

Visualization of conductive information on the OLEB was also investigated with the physical and transient contact of a metallic substance on the board. A thin micropatterned Cr/Au film was developed on a topographic pre-patterned poly(dimethyl siloxane) (PDMS) substrate with periodic one-dimensional lines of 5 and 10 μm in width and periodicity, respectively, followed by the conformal contact of the metallic pattern onto the board (Fig. 2g). The board immediately emitted light upon the physical contact of the pattern (Fig. 2h), with the EL clearly corresponding to the Cr/Au pattern. We noted that the EL pattern is not uniform in light intensity due to non-homogeneous contact of the metallic lines on the board. It should be noted that although numerous AC EL devices have been demonstrated since the first study in 1974 (ref. 23), the visualization of conductance on a parallel AC EL device is unique, making it suitable for imaging fingerprints, as described next.

**Fingerprint detection and imaging.** The unique light-emitting principle of our OLEB allows us to develop a fluorescent image of a human fingerprint when a finger is gently touched to the SiO₂ insulating layer as per Fig. 3a. Considering that the sheet resistance of a human finger ranges from ∼300 to 1,000 Ω sq⁻¹ which corresponds to ∼200–400 S cm⁻¹ (Supplementary Fig. 12), a finger is sufficiently low in sheet resistance for being imaged on our OLEB. A clearly distinguishable fingerprint pattern appeared when the device was touched, with an applied voltage and frequency of 30 V and 100 kHz, as shown in Fig. 3b. The light intensity of the fingerprint again increases with the applied voltage, as characterized by the increase of current as per Fig. 3c, and the fingerprint becomes too bright to be recognizable at voltages higher than 30 V; a voltage of 20 V is sufficient for a readable print. The pattern formed instantaneously, and the device current increased with applied frequency (Supplementary Fig. 14). The current instantly increases at every touch event with a response time of 100 ms, as shown in Fig. 3d. Finger contact on the SiO₂ insulating layer was reliably detected and there was no significant current degradation even after > 20 touch events (Fig. 3e). In addition, contaminants from the fingers were readily removed by cleaning the surface with solvents such as ethanol

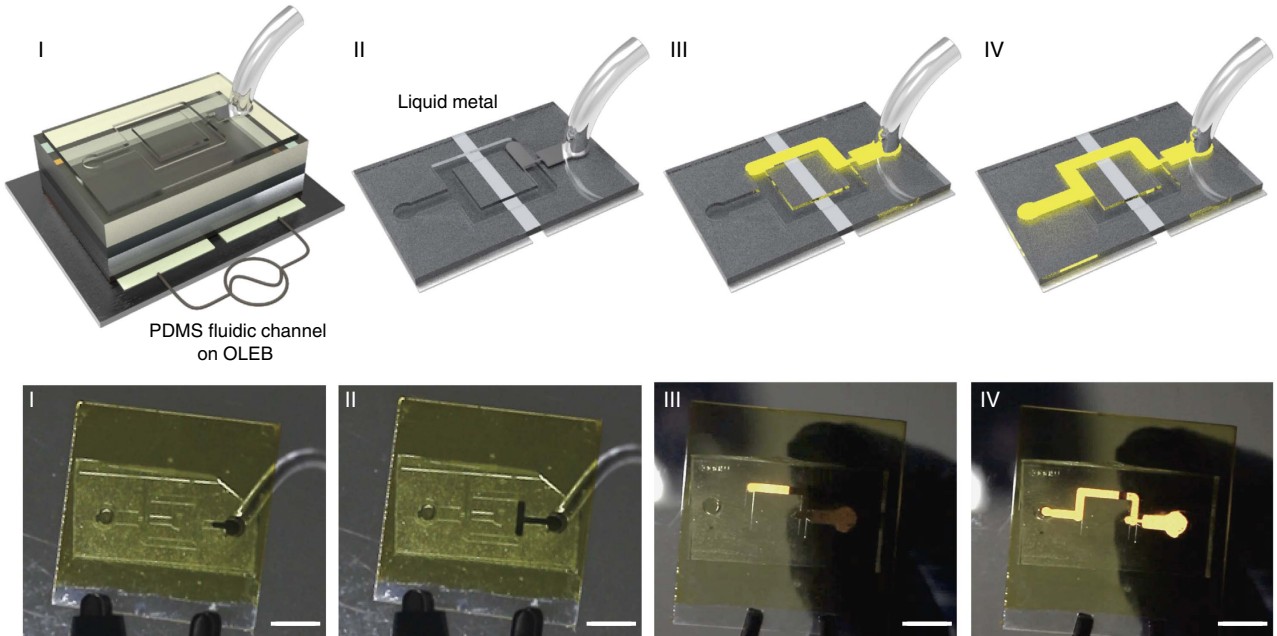

**Figure 4 | Direct imaging of dynamic metallic fluid on OLEB.** Captured photographs of monitoring dynamic flow of a eutectic Gallium-Indium liquid metal fluid through fluidic channel made of poly(dimethyl siloxane) (PDMS) on an organic light emitting board (OLEB). Schematics of the corresponding steps are also shown above the photographs. EL was not observed before injected fluid crossed over the underlying gap between two Indium tin oxide (ITO) parallel electrodes. (Step I and II) The light was turned on as soon as the fluid crossed over the gap and the advancing metallic fluid was clearly visualized with time. (Step III and IV) When a fluid was completely filled in the channel, the channel was fully visualized with the small break arising from the gap between the two parallel ITO electrodes. (Step IV) The alternating-current (AC) frequency was applied of 100 kHz to the device. The scale bar is 5 mm.

and acetone, and therefore can be imaged repeatedly and reliably. Imaging fingerprints with conventional OLEDs is difficult because high-density pixel arrays of OLEDs are required that are individually addressable on pressure or touch-sensitive electric signals. The high-density pixel arrays required for precise recognition make this approach extremely expensive. Our OLEB can be considered as an interactive display because instant current sensing occurred on a touch event with direct visualization of the fingerprint. We also performed the stability test of an OLEB and the results show that the brightness of $10 \, \mathrm{cd} \, \mathrm{m}^{-2}$, which corresponded to $\sim 70\%$ of the initial brightness of $15 \, \mathrm{cd} \, \mathrm{m}^{-2}$ was maintained after 15 h (Supplementary Fig. 15). The luminance is sufficiently high for fingerprint detection. Further improvement on device stability can be achieved by optimized passivation.

Fingerprint imaging was also performed with parallel devices containing green and blue emitting polymers, as shown in Fig. 3f. It should be noted that the operation voltage and frequency were varied for optimum pattern generation because of the difference in energy levels of the fluorescent polymers. Moreover, the OLEB was mechanically flexible without significant performance degradation when fabricated on a plastic poly(ethylene terephthalate) substrate (Fig. 3g). The luminance of the flexible device did not vary with bending radius and a light intensity of several hundred $\mathrm{cd} \, \mathrm{m}^{-2}$ was successfully maintained with a bending radius of 0.5 mm (Supplementary Fig. 16). For further examination of the mechanical flexibility of our device, we also monitored the device performance under the multiple bending cycles and the results show that negligible degradation of the performance was observed up to 1,000 cycles (Supplementary Fig. 16c) The highly bendable device successfully conformed to contacting finger, giving rise to light emission (right, Fig. 3g). This straightforward fingerprint imaging suggests that patch-type ultra-flexible pattern recognition based on polymer EL can be achieved with material optimization. It should be noted that fingerprint detection was

obtained with an applied voltage of 15 V and frequency of 100 kHz with a 100-nm-thick $SiO_2$ insulator (Supplementary Fig. 10).

Positional information of where a finger touched the board was obtained with an OLEB containing arrays of parallel-type ITO electrodes (Fig. 3h). Sets of parallel-type ITO electrodes ($3 \times 3$) were fabricated, onto which OLEB layers were deposited, including ZnO, PEI, a light-emitting layer and $SiO_2$. We examined the light emission arising from finger touch, and the location where a finger was touched was clearly visualized, as shown in Fig. 3i. The early stage fingerprint sensors use cameras, which can pick up the reflective image of a fingerprint produced with LED source[42]. To overcome the problems of the optical sensors including image quality, compactness and cost[43,44], the recent technology indirectly visualizes a fingerprint based on the position selective change in capacitance, temperature and pressure on finger contact which is in turn converted to fingerprint images[45]. Not to mention the indirect imaging of a fingerprint, high-density pixel arrays required for the precise recognition make this approach still expensive. Although several methods have been demonstrated to directly detect fingerprints using radio frequency[44] and acoustic(ultrasound)[46], only a few works deal with direct imaging of fingerprints based on organic and/or polymeric EL. It should be, however, noteworthy that in spite of the successful visualization of a fingerprint even without pixelated display arrays, quantitative sensing of conductance is still limited to the low-conductance regime, as described earlier.

**Direct dynamic imaging of metallic fluid**. The capability of visualizing conductive information can be extended to dynamic monitoring of metallic fluid moving on the board. To demonstrate this we built a fluidic channel made of PDMS on an OLEB, as shown in Fig. 4. A bifurcated fluidic channel of 500-μm-width

was fabricated by photolithography, followed by the replica fabrication with PDMS. Circular inlets and outlets were fabricated (Supplementary Fig. 17). We chose eutectic Ga–In as the conductive fluid. No light was emitted before the fluid flowed over the gap between the parallel ITO electrodes, as shown in steps I and II of Fig. 4. Bright EL was obtained as soon as the fluid crossed the gap (step III), and the advancing metallic fluid was clearly visualized. When the fluid reached the outlet, the EL displayed the pattern of metallic fluid in the channel, as shown in step IV of Fig. 4. Considering that our OLEB is very in early stage of the development, there are numerous issues for further development such as scaling up from a single pixel to higher resolution, full colour operation of our OLEB and direct visualization of biological fluids on OLEB.

## Discussion

We demonstrated a single pixel OLEB device suitable for directly visualizing conductive symbols, patterns and characters written on the board. Our OLEB was based on a unique architecture of stacked layers of $SiO_2$, PEDOT:PSS, a light-emitting layer and ZnO/PEI on two transparent ITO electrodes separated with an in-plane gap. Light emission occurred at the two overlapped regions when a metal without electrical contact was placed on the $SiO_2$ board geometrically across the in-plane ITO electrodes under alternating-current field. The top metallic materials of the OLEB can be chosen regardless of their work functions, making our device useful for visualizing even fingerprint patterns when the $SiO_2$ layer is touched with a naturally conductive human finger. Reliable sensing as well as pattern recognition was simultaneously demonstrated even after multiple touch events owing to the top $SiO_2$ layer efficiently encapsulating the polymeric emitting layer. Highly conformal finger contact was achieved on an OLEB fabricated on a mechanically flexible and optically transparent substrate. Furthermore, we demonstrated visualizing the aforementioned static information and monitoring the dynamic flow of a metallic fluid. The development of this OLEB is extremely useful for simultaneously sensing and directly visualizing conductive information, and is a step forward for the realization of dynamic interactive displays.

## Methods

**Materials.** A super yellow fluorescent polymer (Product: PDY-132), green (product PDO-134), and blue poly(spirobifluorene)-based copolymers (product SPB-02T) were purchased from Merck Co. MWNTs (Grade:TMC220-10) grown by CVD and purified over 95 wt% were manufactured by Nano Solution, Inc., Seoul, Korea. Poly(styrene-*block*-vinyl pyridine) (PS-*b*-P4VP), a dispersant of MWNTs was synthesized by Polymer Source, Inc., Doval, Canada. The PEDOT:PSS (Clevios P VP AI4,083) was modified by being mixed with 0.5 wt% Zonyl surfactant (FS-300 fluorosurfactant from Aldrich) with respect to PEDOT:PSS which promoted the wetting of the PEDOT:PSS layer on an emission layer. High-conductivity PEDOT:PSS (Clevios PH1,000) was also modified by being mixed with 6 wt% of dimethyl sulfoxide (DMSO) and 0.5 wt% Zonyl surfactant with respect to PEDOT:PSS. Zinc acetylacetonate hydrate ($Zn(acac)_2$) and PEI were purchased from Sigma-Aldrich. All other materials were purchased from Aldrich and used as received.

**Fabrication of field-induced OLEB.** A field-induced OLEB was developed with parallel AC EL type device architecture as illustrated in Fig. 1a. First, two in-plane bottom indium tin oxide (ITO) electrodes with the thickness and sheet resistance of 80 nm and $30 \Omega \, cm^{-2}$, respectively, were sputtered onto a glass substrate, with a metal mask used for electrode patterning. The glass substrate with two ITO electrodes was sequentially cleaned twice, before subsequent deposition, with acetone and 2-propanol in an ultrasonic bath for 15 min each, followed by ultraviolet treatment for 15 min. Zinc acetylacetonate hydrate ($Zn(AcAc)_2$) was dissolved in anhydrous ethanol ($25 \, mg \, ml^{-1}$) and stirred at 50 °C for 24 h. The precursor solution was subsequently filtered through a poly(tetrafluoro ethylene) filter (pore diameter: 0.45 μm) to remove agglomerates of undissolved precursor. A ZnO precursor solution was spin-coated onto the cleaned ITO-coated glass, followed by thermal annealing in ambient atmosphere at 120 °C for 30 s, giving rise to a uniform 10-nm-thick ZnO film. Subsequently, 0.4 wt% PEI dissolved in

2-methoxyethanol was spin-coated onto the ZnO film, followed by thermal annealing at 100 °C for 10 min in ambient atmosphere. A Super Yellow emissive film was prepared by spin-coating a solution which had been blended with a dispersion of MWNTs in toluene (0.5 wt%) onto the PEI/ZnO layers. MWNTs were efficiently dispersed by PS-*b*-P4VP and used for enhancing charge injection efficiency[24,25]. A PEDOT:PSS layer was then spin-coated from the modified solution, as described above. A thin $SiO_2$ layer was subsequently sputtered onto the PEDOT:PSS film in high-vacuum ($10^{-6}$ torr). A variety of top metal electrodes such as Al, Ag, Cu and Au were deposited by thermal evaporation. In addition to the thermally evaporated metal electrodes, conductive PEDOT:PSS ink and silver paste were also employed as top electrodes on the board for writing metallic information. A bifurcated fluidic channel of 500-μm-width was developed by conventional photolithography, followed by replication of the channel with PDMS. Circular type inlet and outlet were also fabricated for fluid flow.

**Characterization methods.** The surface morphology of the constituent layers was analysed using tapping-mode AFM (Nanoscope Iva, Digital Instruments) in height and phase contrast. Cross-sectional view of the device was obtained using focused-ion-beam transmission electron microscopy (FIBTEM) (JIB-4,601F, JEOL). The luminance and EL spectra of the devices were obtained using a spectroradiometer (Konica CS 2,000). The current-voltage-luminance (*I-V-L*) characteristics of the devices were measured with a multichannel precision alternating-current power analyzer (ZIMMER Electronics Systems LMG 500). All measurements were performed in a dark box under ambient conditions in air.

**Data availability.** The data that support the findings of this study are available from the corresponding author on request.

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

## Acknowledgements

This research was supported by a grant from the National Research Foundation of Korea (NRF), funded by Korean government (MEST) (No. 2014R1A2A1A01005046). This work was also supported by the LG display and by the third stage of the Brain Korea 21 Plus project in 2014.

## Author contributions

C.P., W.S. E.H.K. and S.H.C. conceived and designed the experiments. E.H.K. performed the fabrication and device characterization of an OLEB. S.H.C. and J.H.L. and B.J. performed the experiments on writable OLEB and fingerprint imaging on OLEB. R.H.K. and S.Y. prepared the figures. C.P. supervised the project, analysed the data and wrote the paper. All authors discussed the results and commented on the manuscript.

## Additional information

**Competing interests:** The authors declare no competing financial interests.

