## [Peer Review File · Nature Communications]

Reviewers' comments:

Reviewer #1 (Remarks to the Author):

Interactive display will be one of the significant display technologies in the future, especially in the application of visualization. The present manuscript is like a popular science article on interactive display, which the key points of this technology and operation principle of OLEB were introduced in detail. The special device structure with two parallel ITO produced an interesting result by an AC electric field. Moreover, the application of fingerprint imaging has a great breakthrough than conventional OLEDs. So, the work of this manuscript is meaningful for personal privacy and public security.

I have some comments for improvement.

For example,

1) "although the optical display elements were based on absorbance and reflective modes, thus limiting lighting efficiency." in which no more explanation for the detail of limiting lighting efficiency.

2) In the instruction of promising OLEDs, the authors have missed out some latest reference.

i.e. J. W. Xu, D. L. Carroll, G. M. Smith, C. C. Dun, Y. Cui, "Achieving High Performance in AC-Field Driven Organic Light Sources", *Scientific Reports*, 6 (2016) 24116.

Y. Xia, Y. Chen, G. M. Smith, D. L. Carroll, "Effects of Electrode Modification Using Calcium on The Performance of Alternating Current Field-Induced Polymer Electroluminescent Devices", *Appl. Phys. Lett.*, 102 (2013) 253302.

Y. Chen, G. M. Smith, E. Loughman, Y. Li, W. Nie, and D. L. Carroll, "Effect of Multi-Walled Carbon Nanotubes on Electron Injection and Charge Generation in AC Field-Induced Polymer Electroluminescence", *Org. Electronics.*, 14 (2013) 8-18.

3) In the discussion of fingerprint detection and imaging, the content is too thin and lack of mechanism analysis. Just give several separate examples.

4) We know the stability of device is very important for the actual life and production applications, authors should pay attention to emphasize this inevitable point with impressive results.

The manuscript is well written and has a clear logic line, but still exists the problem above. There is larger room to improve for well receiving.

Reviewer #2 (Remarks to the Author):

This manuscript describes a new type of electroluminescent interactive display that can actively detect and visualize the external conductive object under an alternating field. The device is composed mostly of organic materials and can be built on flexible substrate to achieve a compliant interface. The authors well explored material characterizations, device configurations, and electrical properties of the device and performed various demonstrations to illustrate its functions.

I recommend acceptance after the authors consider the following revisions:

- Page 2, line 31: Please explain "natural conductivity" of a human fingerprint.
- Page 4, line 66: Please explicitly discuss why the architecture of the insulating layers can readily address band gap issues between the electrodes and light emitting layers and how is this issue critical to the performance of the device.
- Page 4, line 77: The phrase "work functions" may not be clear to a general readership. The authors may consider switch it to conductivity or other terms.
- Page 5, line 103: The authors did not explain what is "Super Yellow". Please give the full name (chemical name) before using it.
- Page 7, line 140: The authors need to add a reference for "light intensity of each LEU increases

with the voltage”.

- Figure 1d: the data for Luminance vs. Voltage at 400kHz is missing in the high voltage part. The authors should add data for that section in order to make the conclusion that 400kHz yields poor light emission. This also applies to Supplementary Figure 6.

- Page 8, line 174: Please discuss how much higher the voltage is applied before the gap in fluorescent image disappears.

- In Summary: Please add discussion for limitations of this technology for the next step. For example, what is the challenge of scaling up from a single pixel to higher resolution; is it possible to incorporate multiple colors of display into one board; how can this visualizing mechanism monitor biological fluids in detail.

Reviewer #3 (Remarks to the Author):

This manuscript reports an investigation on organic light emitting board for dynamic interactive display, which is not suitable for publication in the prestigious journal, such as Nature Communications based on the reasons given below.

1) User-interactive organic electroluminescent skins for instantaneous pressure visualization have been reported previously. (Nature Mater. 899, 12(2013); Science 1071, 351(2016)). The reported performance and functionalities are even better than the current manuscript, such as high stretchability. Therefore the novelty of this study is weak to justify for publication in a good journal.

2) The working principle based on field-induced AC EL platform can also be seen in published report. (Science 1071, 351(2016))

3) One of the drawbacks of the designed structure is the appearance of the gap in the imaging patterns, which arises from the two parallel ITO electrodes. Even though the gap may be able to be eliminated under a high external voltage. However, the power consumption becomes high.

4) The mechanical flexibility of the devices is poorly demonstrated without showing the repeating cycles. It is known that inorganic materials, such as ZnO and ITO films used in the current devices, are not good candidates for flexible and stretchable devices. Thus, it is expected that the devices shown in this manuscript have the difficulty for flexible and stretchable applications.

5) Because this manuscript still provides an alternative structure for user-interactive electroluminescence devices for pressure visualization, it may be more suitable for publication in materials-oriented journals.

Reviewer #4 (Remarks to the Author):

In the manuscript by Kim et al. entitled “Organic Light Emitting Board for Dynamic Interactive Display,” the authors demonstrate a curious device structure which, by means of a floating top electrode, allowing for instantaneous visualization of conductive information present atop device; whether it be an electrode, conductive ink, or even a human fingerprint.

Overall, I thought that this manuscript was interesting, since it employed a rather novel concept to achieve spatial sensing of some conductive stimuli. Unfortunately, due to its insensitivity to the electronic nature of the stimulus atop SiO₂, the device also loses its ability to quantify anything; a large drawback. The operation voltage of the device is high (>10 V) but the concept is interesting. The temporal response is also ~ 100 ms, which seems satisfactory for the applications mentioned

in the manuscript.

I'd like to recommend this manuscript for publication, provided the following questions and comments are addressed:

(1) As a justification for the author's approach, it was stated that previous efforts that also sought to achieve some sort of "stimuli-interfacing" have their shortcomings. Namely, those based on microfluidics, organic electrochromics, thermochromics, and OLEDs coupled with pressure sensors. However, in the current manuscript, such quantitative sensing, (save for spatial distribution of conductivity) of external stimuli cannot be achieved. In this sense, I am not convinced that the introduction of this paper is suitable to describe the conclusion of the paper.

(2) Many of the figures are not quantitatively labeled when they should be. For example, please see, Figure 1c, Figure 2b, Figure 2c, Figure 3e, and Supplementary 4b. Please add the numerical scale. It hard to judge the data properly if it is plotted like this.

(3) Is the use of a SWNT composite really necessary for use in the emitting layer of this device? How does the device work without this composite? In the case of super yellow, the largest injection barrier in the described structure is most likely from the ZnO/PEIE contact.

(4) From the work cited in reference 25, the result showing that higher conductive (more doped) PEDOT:PSS gives better performance is hardly a surprising result. Have you also considered doping the electron injection (ITO/ZnO/PEIE) contact to lower the operation voltage?

Reviewers' comments:

Reviewer #1 (Remarks to the Author):

General Comment: Interactive display will be one of the significant display technologies in the future, especially in the application of visualization. The present manuscript is like a popular science article on interactive display, which the key points of this technology and operation principle of OLEB were introduced in detail. The special device structure with two parallel ITO produced an interesting result by an AC electric field. Moreover, the application of fingerprint imaging has a great breakthrough than conventional OLEDs. So, the work of this manuscript is meaningful for personal privacy and public security. I have some comments for improvement. For example,

Comment 1. “although the optical display elements were based on absorbance and reflective modes, thus limiting lighting efficiency.” in which no more explanation for the detail of limiting lighting efficiency.

→ **Response:** We have meant that the lighting properties of the displays based on either absorbance or reflectivity such as brightness, response time and efficiency is much lower than that on electroluminescence. To avoid the confusion, we modified the sentence as follows;

“although the optical display elements were based on absorbance and reflective modes which in general suffer from rather low brightness, slow response time and low light efficiency.”

Comment 2. In the instruction of promising OLEDs, the authors have missed out some latest reference.

i.e. J. W. Xu, D. L. Carroll, G. M. Smith, C. C. Dun, Y. Cui, “Achieving High Performance in AC-Field Driven Organic Light Sources”, Scientific Reports, 6 (2016) 24116.

Y. Xia, Y. Chen, G. M. Smith, D. L. Carroll, “Effects of Electrode Modification Using Calcium on The Performance of Alternating Current Field-Induced Polymer Electroluminescent Devices”, Appl. Phys. Lett., 102 (2013) 253302.

Y. Chen, G. M. Smith, E. Loughman, Y. Li, W. Nie, and D. L. Carroll, “Effect of Multi-Walled Carbon Nanotubes on Electron Injection and Charge Generation in AC Field-Induced Polymer Electroluminescence”, Org. Electronics., 14 (2013) 8-18.

→ **Response:** We appreciate the relevant references the Reviewer mentioned. As the Reviewer suggested, we cited these papers.

Comment 3. In the discussion of fingerprint detection and imaging, the content is too thin and lack of mechanism analysis. Just give several separate examples.

→ **Response:** As the Reviewer suggested, we discussed more in details the mechanism analysis for the fingerprint detection from the literature. We provided several separate examples of fingerprint analysis methods with their detection mechanisms as the Reviewer suggested. Furthermore, we demonstrated that the conductivity of finger was sufficiently high and visualized on OLEB by correlating the image with the master relation of conductivity vs.

light emission properties developed with conductivity controlled PEDOT:PSS as shown in the modified Supplementary Figure, 12. The discussion on the mechanism analysis shown below was made in the page 12,13 of the revised manuscript.

“The early stage fingerprint sensors use cameras which can pick up the reflective image of a fingerprint produced with LED source¹. In order to overcome the problems of the optical sensors including image quality, compactness and cost^{2,3}, the recent technology indirectly visualizes a fingerprint based on the position selective change in capacitance, temperature and pressure upon finger contact which is in turn converted to finger print images⁴. Not to mention the indirect imaging of a fingerprint, high density pixel arrays required for the precise recognition make this approach still expensive. Although several methods have been demonstrated to directly detect fingerprints using radio frequency³ and acoustic(ultrasound)⁵, only a few works deal with direct imaging of fingerprints based on organic and/or polymeric EL.”

1. Xia, X., & O'Gorman, L. Innovations in fingerprint capture devices. *Pattern Recognition* **36**, 361-369 (2013).
2. Nam, J. M., Jung, S. M., & Lee, M. K. Design and implementation of a capacitive fingerprint sensor circuit in CMOS technology. *Sensors and Actuators A: Physical* **135**, 283-291 (2007).
3. Alonso-Fernandez, F., et al. Performance of fingerprint quality measures depending on sensor technology. *Journal of Electronic Imaging* **17**, 011008-011008 (2008).
4. Shimoyama, N., et al. Effect of scratch stress on the surface hardness and inner structures of a capacitive fingerprint sensor LSI. *2007 IEEE International Reliability Physics Symposium Proceedings. 45th Annual. IEEE* (2007).
5. Ross, A., & Jain, A., Biometric sensor interoperability: A case study in fingerprints. *International Workshop on Biometric Authentication*. Springer Berlin Heidelberg 134-145 (2004).

Comment 4. We know the stability of device is very important for the actual life and production applications, authors should pay attention to emphasize this inevitable point with impressive results.

→ **Response:** We agree with the Reviewer that the stability of device is important, considering the strict requirement of life time of the commercialized displays. Although AC EL devices are in their early stage of the development, the device stability should be carefully addressed for their potential applications for display. From academic institutions, however, there have been not many works which have dealt with the operation stability based on the experimental protocol used for commercial OLED due to the difficulty in setting up experimental conditions optimized for lifetime studies. We also performed the stability test of an OLEB and the results show that the brightness of 10 cd m⁻² which corresponded to approximately 70% of the initial brightness of 15 cd m⁻² was maintained after 15 hours as shown in a new Supplementary Figure, 15. As noted, the brightness of 10 cd m⁻² is sufficiently high for fingerprint detection. Further improvement on device stability can be achieved by optimized passivation methods well developed in the OLEDs. We discussed this issue in the page 11 of the revised manuscript with the Supporting information.

Reviewer #2 (Remarks to the Author):

General Comment: This manuscript describes a new type of electroluminescent interactive display that can actively detect and visualize the external conductive object under an alternating field. The device is composed mostly of organic materials and can be built on flexible substrate to achieve a compliant interface. The authors well explored material characterizations, device configurations, and electrical properties of the device and performed various demonstrations to illustrate its functions. I recommend acceptance after the authors consider the following revisions:

Comment 1. Page 2, line 31: Please explain “natural conductivity” of a human fingerprint.

→ **Response:** The literature shows that the sheet resistance of a human finger ranges from approximately 300 to 1000 $\Omega/$ which corresponds to approximately 200 to 400 S/cm, based on our results in the Supplementary Figure, 12. The sheet resistance is sufficiently low for being imaged on our OLEB. The measured value of 400 $\Omega/$ in the laboratory was also similar to that in the literature. We discussed this issue in the page 10 of the revised manuscript.

Comment 2. Page 4, line 66: Please explicitly discuss why the architecture of the insulating layers can readily address band gap issues between the electrodes and light emitting layers and how is this issue critical to the performance of the device.

→ **Response:** We have meant that our architecture allows for work function independent operation of a metallic object on top of an insulating layer but for conductivity dependent operation of the metal. As the Reviewer is aware, work function of a metal becomes important when the metal is on contact with another metal or semiconductor, which is not the case in our work where two layers are separated with an insulating layer. Our results in Supplementary Figure, 12 show that the light emission properties enhance with the conductivity of the materials up to 700 S cm^{-1} on top of an insulating layer. The properties were, however, unaltered with conventional metals whose conductivity values are higher than 10,000 S cm^{-1} . In this high conductivity regime, the device operation becomes independent of the top electrode. In other words, both conductivity and work function do not have to be considered. We discussed this issue in details in the page 9 of the revised manuscript.

Comment 3. Page 4, line 77: The phrase “work functions” may not be clear to a general readership. The authors may consider switch it to conductivity or other terms.

→ **Response:** Related to the previous answer, we explicitly explained the working principle of our device which allowed for work function independent but conductivity dependent light emitting operation in the page 9 of the revised manuscript.

Comment 4. Page 5, line 103: The authors did not explain what is “Super Yellow”. Please

give the full name (chemical name) before using it.

→ **Response:** We are regretful for missing the full name of one of the organic emitters, super yellow. We provided its full name in the experimental parts.

Comment 5. Page 7, line 140: The authors need to add a reference for “light intensity of each LEU increases with the voltage”.

→ **Response:** As the Reviewer suggested, we included three references where light intensity of an AC EL device increased with the operation voltage.

Comment 6. Figure 1d: the data for Luminance vs. Voltage at 400kHz is missing in the high voltage part. The authors should add data for that section in order to make the conclusion that 400kHz yields poor light emission. This also applies to Supplementary Figure 6.

→ **Response:** In typical AC EL devices, the device performance become degraded when the frequency was above a certain value due to the limited carrier mobility of the injected carriers that was not sufficiently fast to follow the switching speed of bias polarity. The critical frequency value at which performance degradation occurs depends on not only materials composed of the device but also device architecture. In our device, 400 kHz was sufficiently high, giving rise to the poor light emission. At the high operation voltage, the device was electrically broken down. To confirm the device failure at 400 kHz, we included the results of a device operated at 200 and 300 kHz in a new Supplementary Figure, 6. The device performance was degraded at 200 kHz, compared with that at 100 kHz and the failure occurred at approximately 60 V when it was operated both 300 and 400 kHz. We addressed this issue in the page 7 with the Supporting information.

Comment 7. Page 8, line 174: Please discuss how much higher the voltage is applied before the gap in fluorescent image disappears.

→ **Response:** The voltage required for disappearance of the gap in the device mostly depends upon the width of an initial gap. In the writable board demonstrated with the gap of 0.5 mm, the gap looks disappearing with the voltage of approximately 70 V. In order to have more quantitative relation between gap and critical voltage, we fabricated several parallel AC EL devices with different gaps of 1mm, 0.5mm, 0.1mm, 0.05mm, 0.025mm, 0.01mm by conventional photolithography, followed by metal deposition and lift-off as shown in a new Supplementary Figure, 13. As shown in the results, the gap became invisible with its width of 50 μm and below when the device was operated at the constant voltage of 30 V which was close to the turn-on voltage of the device. The results clearly show that the gap in our parallel AC EL device is not a critical issue but a minor drawback for further development. We explicitly addressed this issue in the page 9,10 with the Supporting information.

Comment 8.- In Summary: Please add discussion for limitations of this technology for the

next step. For example, what is the challenge of scaling up from a single pixel to higher resolution; is it possible to incorporate multiple colors of display into one board; how can this visualizing mechanism monitor biological fluids in detail.

→ **Response:** Our OLEB is very in early stage of the development. There are numerous challenges for further development which include how efficiently to isolate the AC electric field from one device to another when scaling up from a single pixel to higher resolution, how to realize full color operation of our OLEB and how to fabricate conductive medium containing biological objects for direct visualization of biological fluids on OLEB as the Reviewer thankfully mentioned. We briefly discussed these issues more in details in the discussion part of the page 13.

Reviewer #3 (Remarks to the Author):

General Comment: This manuscript reports an investigation on organic light emitting board for dynamic interactive display, which is not suitable for publication in the prestigious journal, such as Nature Communications based on the reasons given below.

Comment 1. User-interactive organic electroluminescent skins for instantaneous pressure visualization have been reported previously.(Nature Mater. 899, 12(2013); Science 1071, 351(2016). The reported performance and functionalities are even better than the current manuscript, such as high stretchability. Therefore the novelty of this study is weak to justify for publication in a good journal.

→ **Response:** We appreciated the Reviewer's comments. Although we respect the Reviewer's comments, we are not able to agree with the Reviewer's comments with the following reasons. User-interactive EL based devices have been widely demonstrated including the works the Reviewer mentioned as we summarized in a Supporting information, Table S1. It is not surprising that some previous works of the user interactive organic EL devices exist, considering the importance of the growing research field. In fact, we clearly mentioned these references in the introduction parts with their merits and demerits. Although most of approaches so far in this user-interactive EL field have been realized based "pressure-visualization", we suggest a completely different approach based on "conductance visualization". We assume that the Reviewer seems to misunderstand the core part of the manuscript, judging from not only the Reviewer's comment above but also the Reviewer's question#5 "Because this manuscript still provides an alternative structure for user-interactive electroluminescence devices for "**pressure visualization**", it may be more suitable for publication in materials-oriented journals". Furthermore, Table S1 we prepared shows the comparison of these works with ours in terms of functionality. For instance, while the work for the pressure visualization was built with two device components of OLED and pressure sensing layer in series, our device does not require an additional sensing layer (i.e. conductive layer is one of our EL device component). More importantly, our device is clearly different in working mechanism from the pressure visualization work because ours directly visualized the

conductance rather than pressure. The Science paper the Reviewer mentioned is another excellent work dealing with the user-interactive AC EL combined with pressure sensing arrays which shows surprising mechanical flexibility. As shown in the Table S1, the device requires several kilo volts for field-induced EL operation due to the inorganic light emitting materials embedded in elastic polymer medium for both flexibility and stretchability. On the other hand, our device with stacked thin layers needs a few tens of volts for the operation (2 orders of magnitude lower). Demonstration of stretchable EL devices is not our main focus. Instead, our vertically stacked, multilayered structure similar to conventional OLED architecture is adopted for our OLEB, which is a novel concept of organic light-emitting displays. The mechanical flexibility under bending in OLEDs on flexible substrate has been achieved widely and thus we also demonstrated that our OLEB was flexible up to the bending radius of approximately 5 mm (Supplementary Figure 16). The mechanical flexibility of our device is very comparable with that of devices reported in OLEDs. The advantages of our OLEB with the unique capability of directly visualizing conductance on a board clearly differentiate our work from others, rendering the other Reviewers recognize the novelty of our work and agree that the work can be suitable for publication after addressing the valuable issues they raised for further improvement of our work. Although we hardly agree with the Reviewer's comments, we modified our manuscript to avoid the possible misunderstandings with a new Table S1.

Comment 2. The working principle based on field-induced AC EL platform can also be seen in published report.(Science 1071, 351(2016))

➔ **Response:** AC EL has been of great interest since its first report at 1974 from Sharp incorporation, Japan and numerous works have appeared not only to improve device performance but also to elucidate its unique working principle. Our group has also contributed to developing AC EL devices, in particular, based on ones with organic light emitters, including light emitting composites with networked carbon nanotubes[Nano Lett. 11, 966-972(2011)], white AC EL[ACS Nano 7, 10809-10817(2013)], scratch resistive AC EL and full color high luminescent AC EL[J. Polym. Sci., Part B: Polym. Phys. 53, 1629-1640(2015)]. Recently, we also demonstrated that light emission properties were readily controlled in non-volatile manner with a ferroelectric charge injection gate layer under AC operation, giving rise to a novel EL memory where EL states with different intensities were programmed, read and erased repetitively[Adv. Funct. Mater. 26, 5391-5399(2016)]. It is, therefore, hardly surprising that the working principle of AC EL is seen in the Science paper. Again, our work adopts its device architecture with in-plane, parallel AC field which allows for direct visualization of conductance on the novel OLEB. As previously mentioned, we scrutinized and summarized the AC EL devices sharing the working principle with ours in Table S1. In addition, we discussed this issue in the page 10 of the revised manuscript.

Comment 3. One of the drawbacks of the designed structure is the appearance of the gap in the imaging patterns, which arises from the two parallel ITO electrodes. Even though the gap may be able to be eliminated under a high external voltage. However, the power consumption becomes high.

→ **Response:** We agree that the gap in our OLEB may be a drawback for writing process as we clearly mentioned in the manuscript. We also pointed out that the gap can be invisible with a sufficient voltage. The voltage required for disappearance of the gap in the device mostly depends upon the width of an initial gap. In the writable board demonstrated with the gap of 0.5 mm, the gap looks disappearing with the voltage of approximately 70 V. In order to have more quantitative relation between gap and critical voltage, we fabricated several parallel AC EL devices with different gaps of 1mm, 0.5mm, 0.1mm, 0.05mm, 0.025mm, 0.01mm by conventional photolithography, followed by metal deposition and lift-off as shown in a new Supplementary Figure, 13. As shown in the results, the gap became invisible with its width of 50 μm and below when the device was operated at the constant voltage of 30 V which was close to the turn-on voltage of the device. The power consumption for the gap disappearance is not high, in particular, compared with that for the stretchable AC EL the Reviewer mentioned. The results clearly show that the gap in our parallel AC EL device is not a critical issue but a minor drawback for further development. We explicitly addressed this issue in the page 9,10 with the Supporting information.

Comment 4. The mechanical flexibility of the devices is poorly demonstrated without showing the repeating cycles. It is known that inorganic materials, such as ZnO and ITO films used in the current devices, are not good candidates for flexible and stretchable devices. Thus, it is expected that the devices shown in this manuscript have the difficulty for flexible and stretchable applications.

→ **Response:** As explained above, the demonstration of stretchable EL devices is not our main focus in our AC EL device for display board which follows its structure similar to that of a conventional OLED with vertically stacked multilayers. Nonetheless, we also demonstrated that our OLEB was flexible up to the bending radius of approximately 5 mm (Supplementary Figure 16). The mechanical flexibility of our device is very comparable with that of devices reported in OLEDs. For further examination of the mechanical flexibility of our device, we also monitored the device performance under the multiple bending cycles and the results show that negligible degradation of the performance was observed up to 1000 cycles, which we believe quite remarkable. (Please see the modified Supplementary Figure, 16) We would also like to mention that in order to make an AC EL device stretchable as shown in the literature, light emitting materials should be embedded in the insulating and elastomeric polymer matrix which requires high electric field for excitation of the light emitters and subsequent recombination of excitons, giving rise to a device with its operation voltage of a few kilo volts. We clearly demonstrated that our OLED was mechanically flexible by including the bending cycle results as the Reviewer suggested. We discussed this issue in the page 12 of the revised manuscript.

Comment 5. Because this manuscript still provides an alternative structure for user-interactive electroluminescence devices for pressure visualization, it may be more suitable for publication in materials-oriented journals.

→ **Response:** Our work *does not* visualize pressure but visualize conductance. The Reviewer may misunderstand the concept of our OLEB. We kindly ask the Reviewer to re-evaluate our revised manuscript where we more clearly described the unique function of our OLEB for direct visualization of conductance.

Reviewer #4 (Remarks to the Author):

General Comment: In the manuscript by Kim et al. entitled “Organic Light Emitting Board for Dynamic Interactive Display,” the authors demonstrate a curious device structure which, by means of a floating top electrode, allowing for instantaneous visualization of conductive information present atop device; whether it be an electrode, conductive ink, or even a human fingerprint.

Overall, I thought that this manuscript was interesting, since it employed a rather novel concept to achieve spatial sensing of some conductive stimuli. Unfortunately, due to its insensitivity to the electronic nature of the stimulus atop SiO₂, the device also loses its ability to quantify anything; a large drawback. The operation voltage of the device is high (>10 V) but the concept is interesting. The temporal response is also ~ 100 ms, which seems satisfactory for the applications mentioned in the manuscript.

I'd like to recommend this manuscript for publication, provided the following questions and comments are addressed:

Comment 1. As a justification for the author's approach, it was stated that previous efforts that also sought to achieve some sort of “stimuli-interfacing” have their shortcomings. Namely, those based on microfluidics, organic electrochromics, thermochromics, and OLEDs coupled with pressure sensors. However, in the current manuscript, such quantitative sensing, (save for spatial distribution of conductivity) of external stimuli cannot be achieved. In this sense, I am not convinced that the introduction of this paper is suitable to describe the conclusion of the paper.

→ **Response:** We have not intended to claim that the previous works based on microfluidics, organic electrochromics, thermochromics and OLEDs have their shortcomings to highlight our work in the introduction. We have meant that those devices are excellent candidates for emerging user-interactive displays which have been of great interest in the recent research development. To highlight a new working principle of our OLEB for the convenient spatial visualization of conductive information, we stated that the work based on OLEDs combined with pressure sensing elements requires additional arrays of the devices to visualize the spatial pressure information. In addition, we stated the importance of developing a device which can respond instantaneously to other stimuli, *i.e.* conductance rather than pressure and directly visualize the spatial information of the conductance. We agree in part with the Reviewer that our OLEB is insensitive to electronic nature of materials on top of an insulator and thus lose its ability to quantify the electronic properties. Our results shown in Supplementary Figure, 12 exhibit that the device properties such as light intensity and current

are proportional to the conductivity up to 700 S cm^{-1} on top of an insulating layer. The properties were, however, unaltered with conventional metals whose conductivity values are higher than $10,000 \text{ S cm}^{-1}$. In this high conductivity regime, the device operation becomes insensitive to the top electrode as the Reviewer pointed out. Besides the ability for directly visualizing the spatial information of highly conductive materials regardless of the nature electronic properties of the materials, our OLEB is able to quantitatively address the degree of conductivity which is correlated with the light emitting properties with the materials having certain range of the conductivities. We modified the introduction part by describing more clearly the importance of the previous user-interactive displays as well as by emphasizing the spatial visualization of conductance based on our OLEB which is still limited to the quantitative sensing of the broad range conductance with the modified Supplementary Figure, 12.

Comment 2. Many of the figures are not quantitatively labeled when they should be. For example, please see, Figure 1c, Figure 2b, Figure 2c, Figure 3e, and Supplementary 4b. Please add the numerical scale. It hard to judge the data properly if it is plotted like this.

→ **Response:** We are regretful that we missed the quantitative labels of some of the results. As the Reviewer suggested, we included the numerical scales in the figures the Reviewer mentioned.

Comment 3. Is the use of a SWNT composite really necessary for use in the emitting layer of this device? How does the device work without this composite? In the case of super yellow, the largest injection barrier in the described structure is most likely from the ZnO/PEIE contact.

→ **Response:** Our previous works with vertically stacked AC EL devices show that SWNTs uniformly dispersed in an emission layer have significantly improved device performance by lowering injection barrier of field induced carriers. The light intensity increased with the amount of nanotubes up to a certain concentration. The degradation of the performance was observed with an emission layer containing high SWNT content mainly due to the aggregates of nanotubes at the high concentration which acted as exciton quenchers. (Please refer to the reference 24 in the manuscript) To confirm the role of SWNTs in our parallel AC EL device with the inverted carrier injection structure, we also fabricated a parallel AC EL device without SWNTs. The device shows the performance much worse than one with nanotubes as shown in a new Supplementary Figure, 8. The turn-on voltage was significantly increased with the maximum luminescence much lower than that with SWNTs. As the Reviewer pointed out, we introduced a PEI layer whose strong dipole moment was able to reduce the large injection barrier between ZnO and an emitting layer. We do believe that the barrier was further reduced by SWNTs in the emission layer as explained above. We discussed this issue in the page 7,8 of the revised manuscript with the Supplementary Figure, 8.

Comment 4. From the work cited in reference 25, the result showing that higher conductive

(more doped) PEDOT:PSS gives better performance is hardly a surprising result. Have you also considered doping the electron injection (ITO/ZnO/PEIE) contact to lower the operation voltage?

→ **Response:** We appreciate the Reviewer's suggestion of doping the electron injection contact to lower the operation voltage. In OLEDs, in fact, Li doping in ZnO layer is common to reduce the electron injection barrier, giving rise to the low voltage operation. We examined the light emission performance with a parallel AC EL device in which ZnO layer was doped with Li. In our ZnO/PEI layer solution-processed on the ITO electrode, however, the proper conditions for ion doping were not readily made, giving rise to the operation turn-on voltage a little higher than that without doping. We briefly mentioned the doping effect in the page 7 of the revised manuscript.

To address the Reviewer's concern on the high operation voltage, on the other hand, we also fabricated a parallel AC-EL device with ca. 100 nm thick SiO₂ layer. The turn-on voltage of the device was approximately 10 V as shown in a new Supplementary Figure, 10. The results suggest that our OLEB can be operated at low voltage with device optimization. We addressed this issue in the page 8 of the revised manuscript with the Supplementary Figure, 10.

Reviewers' comments:

Reviewer #1 (Remarks to the Author):

All comments have been addressed. The manuscript can be published as it is.

Reviewer #2 (Remarks to the Author):

In this revised manuscript, the authors have clarified certain points about the functioning mechanisms and various characterizations they performed on their device. Although alternating current electroluminescence is not a novel technology, the authors realized an OLEB that can simultaneously sense and visualize conductive objects by making use of multiple organic materials and designing unique device structures. The fingerprint detection/imaging is certainly a highlight since it demonstrates convincing resolution without the use of high density display arrays. They have well addressed the questions I brought up during the first round of reviews and I recommend acceptance without further revisions. I believe this work will be potentially of interest to researchers in the fields of flexible electronics and optics.

Reviewer #3 (Remarks to the Author):

We thank the authors for carefully answering the comments point-by-point. However, there are still several main concerns about the current manuscript as listed below:

- 1) The application of fingerprint imaging based on OLED is not a great breakthrough, which is well-known in many previous reports.(patent US 20140036168; patent US 20150331508)
- 2) Due to its insensitivity to the electronic nature of the stimulus of SiO₂ layer, the device loses its ability to have quantitative evaluation, which is a major drawback.
- 3) As stated in the revised manuscript, the stability of the current devices still remains as a critical issue, which is far away from practical application.
- 4) The authors emphasized that their devices are based on conductance visualization instead of pressure visualization. However, for most pressure sensors reported previously, their underlying mechanism is to transfer the magnitude of external pressure into the measured current. Therefore, the difference is not very significant.
- 5) User-interactive organic electroluminescent skins for instantaneous touch visualization have been reported previously.(Nature Mater. 899, 12(2013); Science 1071, 351(2016). The reported performance and functionalities are even better than the current manuscript, such as high stretchability. Therefore the novelty of this study is weak to justify for publication in a good journal.
- 6)The working principle based on field-induced AC EL platform can also be seen in published report.(Science 1071, 351(2016)

Reviewer #4 (Remarks to the Author):

I appreciate the authors for taking the time to answer all the reviewer's questions. Below I will reply to the responses to my original review.

Comment #1: So with this kind of "calibration" curve you can get quantitative and spatial sensing of the conductivity of the top surface? Did I understand correctly? Do you think that there's a better type of demonstration to show this capability?

Comment #2: Thank you for updating the labels of the plots.

Comment #3: Thank you for the experiment to help clarify the influence of SWNTs to your current-voltage characteristics.

Comment #4: Thank you for the additional experiment to demonstrate lower voltage operation. Shouldn't this device then be the main feature of your manuscript? I understand you've already collected a lot of data with the higher voltage operation device. Is it difficult to obtain a suitable device yield with a 100 nm thick SiO₂ layer?

Reviewers' comments:

Reviewer #1 (Remarks to the Author):

All comments have been addressed. The manuscript can be published as it is.

Response: We thank Reviewer #1 for previous comments that strengthened the manuscript. We are delighted that Reviewer #1 believes that the manuscript can be accepted in the current manuscript.

Reviewer #2 (Remarks to the Author):

In this revised manuscript, the authors have clarified certain points about the functioning mechanisms and various characterizations they performed on their device. Although alternating current electroluminescence is not a novel technology, the authors realized an OLEB that can simultaneously sense and visualize conductive objects by making use of multiple organic materials and designing unique device structures. The fingerprint detection/imaging is certainly a highlight since it demonstrates convincing resolution without the use of high density display arrays. They have well addressed the questions I brought up during the first round of reviews and I recommend acceptance without further revisions. I believe this work will be potentially of interest to researchers in the fields of flexible electronics and optics.

Response: We thank Reviewer #2 for previous comments that clarified the manuscript. We are delighted that Reviewer #2 believes that the manuscript can be accepted in the current manuscript.

Reviewer #3 (Remarks to the Author):

We thank the authors for carefully answering the comments point-by-point. However, there are still several main concerns about the current manuscript as listed below:

1) The application of fingerprint imaging based on OLED is not a great breakthrough, which is well-known in many previous reports.(patent US 20140036168; patent US 20150331508)

Response: As shown in the previous revised manuscript, we have explicitly discussed in details the mechanism analysis for the fingerprint detection from the literature. We have also provided several separate examples of fingerprint analysis methods with their detection mechanisms. We have clearly mentioned that finger print detection based on OLEDs is one of them in the page 13 in the revised manuscript.

2) Due to its insensitivity to the electronic nature of the stimulus of SiO₂ layer, the device loses its ability to have quantitative evaluation, which is a major drawback.

Response: This issue has been explicitly addressed in our previous revised manuscript in

response to the comment raised by the Reviewer #3. Our results, shown in Supplementary Figure 12, show that the device properties such as light intensity and current are proportional to the conductivity up to 700 S cm^{-1} , allowing for the quantitative evaluation of the conductance. Besides the ability for directly visualizing the spatial information of highly conductive materials, our OLEB is capable of quantitatively addressing the correlated conductivity-light emitting properties.

3) As stated in the revised manuscript, the stability of the current devices still remains as a critical issue, which is far away from practical application.

Response: We agree with the Reviewer that the stability of device is important, considering the strict requirement for long life time of the commercialized displays. Although AC EL devices are in their early stage of the development, the device stability should be carefully addressed for their potential applications for display. From academic institutions, however, there have been not many works which have dealt with the operation stability based on the experimental protocol used for commercial OLED due to the difficulty in setting up experimental conditions optimized for lifetime studies. The stability results of an OLEB show that the brightness of 10 cd m^{-2} which corresponded to approximately 70% of the initial brightness of 15 cd m^{-2} was maintained after 15 hours as shown in the Supplementary Figure, 15. As noted, the brightness of 10 cd m^{-2} is sufficiently high for fingerprint detection. Further improvement on device stability can be achieved by optimized passivation methods well developed in the OLEDs. Please note that the Reviewer 1 who raised the stability issue previously has been satisfied with our revision including the stability results.

4) The authors emphasized that their devices are based on conductance visualization instead of pressure visualization. However, for most pressure sensors reported previously, their underlying mechanism is to transfer the magnitude of external pressure into the measured current. Therefore, the difference is not very significant.

Response: Although most of approaches so far in the field of user-interactive display have been realized in the context of “pressure-visualization”, we suggest a completely different approach based on “conductance visualization”. Specifically, Table S1 shows the comparison of these previous works with ours in terms of functionality. For example, while the work for the pressure visualization was built with two device components of OLED and pressure sensing layer in series, our device does not require an additional sensing layer (i.e. conductive layer is one of our EL device component). More importantly, our device is clearly different in working mechanism from the pressure visualization work because ours directly visualized the conductance rather than pressure. In other words, our device does not require external pressure which would be converted into current. For instance, our device will not work when an insulating object is placed and even pressurized on the board. The light intensity of an OLEB does not depend upon the pressure but upon the conductance of an object on the board, which is, we believe, different from ones based on pressure visualization.

5) User-interactive organic electroluminescent skins for instantaneous touch visualization have been reported previously.(Nature Mater. 899, 12(2013); Science 1071, 351(2016). The reported performance and functionalities are even better than the current manuscript, such as high stretchability. Therefore the novelty of this study is weak to justify for publication in a good journal.

Response: It is regretful that the Reviewer has not been fully convinced with our response which we prepared with our best efforts. We have summarized user-interactive EL devices in the Supporting information, Table S1 in the revised manuscript. Thus, it is not surprising that some previous works exist, considering the importance of the growing research field. In fact, we clearly mentioned these references in the introduction parts with their merits and demerits. Although most of approaches so far in this user-interactive EL field have been realized based “pressure-visualization”, we suggest a completely different approach based on “conductance visualization”. Furthermore, Table S1 we prepared shows the comparison of these works with ours in terms of functionality. For instance, while the work for the pressure visualization was built with two device components of OLED and pressure sensing layer in series, our device does not require an additional sensing layer (i.e. conductive layer is one of our EL device component). More importantly, our device is clearly different in working mechanism from the pressure visualization work because ours directly visualized the conductance rather than pressure. The Science paper the Reviewer mentioned is another excellent work dealing with the user-interactive AC EL combined with pressure sensing arrays which shows surprising mechanical flexibility. As shown in the Table S1, the device requires several kilo volts for field-induced EL operation due to the inorganic light emitting materials embedded in elastic polymer medium for both flexibility and stretchability. On the other hand, our device with stacked thin layers needs a few tens of volts for the operation (2 orders of magnitude lower). Demonstration of stretchable EL devices is not our main focus. Instead, our vertically stacked, multilayered structure similar to conventional OLED architecture is adopted for our OLEB, which is a novel concept of organic light-emitting displays. The mechanical flexibility under bending in OLEDs on flexible substrate has been achieved widely and thus we also demonstrated that our OLEB was flexible up to the bending radius of approximately 5 mm (please see Supplementary Figure 16). The mechanical flexibility of our device is very comparable with that of devices reported in OLEDs. The advantages of our OLEB with the unique capability of directly visualizing conductance on a board clearly differentiate our work from others, rendering the other Reviewers recognize the novelty of our work with approval for publication after the first round revision.

6) The working principle based on field-induced AC EL platform can also be seen in published report.(Science 1071, 351(2016)

Response: It is again regretful that the Reviewer has not been fully convinced with our response which we prepared with our best efforts. AC EL has been of great interest since its first report at 1974 from Sharp incorporation, Japan and numerous works have appeared not only to improve device performance but also to elucidate its unique working principle. Our group has also contributed to developing AC EL devices, in particular, based on ones with organic light emitters, including light emitting composites with networked carbon

nanotubes[Nano Lett. 11, 966-972(2011)], white AC EL[ACS Nano 7, 10809-10817(2013)], scratch resistive AC EL and full color high luminescent AC EL[J. Polym. Sci., Part B: Polym. Phys. 53, 1629-1640(2015)]. Recently, we have also demonstrated that light emission properties were readily controlled in non-volatile manner with a ferroelectric charge injection gate layer under AC operation, giving rise to a novel EL memory where EL states with different intensities were programmed, read and erased repetitively[Adv. Funct. Mater. 26, 5391-5399(2016)]. It is, therefore, hardly surprising that the working principle of AC EL is seen in the Science paper. Again, our work adopts its device architecture with in-plane, parallel AC field which allows for direct visualization of conductance on the novel OLEB. As previously mentioned, we scrutinized and summarized the AC EL devices sharing the working principle with ours in Table S1.

Reviewer #4 (Remarks to the Author):

I appreciate the authors for taking the time to answer all the reviewer's questions. Below I will reply to the responses to my original review.

Comment #1: So with this kind of "calibration" curve you can get quantitative and spatial sensing of the conductivity of the top surface? Did I understand correctly? Do you think that there's a better type of demonstration to show this capability?

Response: In principle, one will be able to obtain the spatial sensing for a conductive object with conductivity in the range up to 700 S cm^{-1} . For example, the same symbols written with PEDOT:PSS inks with different conductivities would exhibit different EL intensity although the spatial information (i.e. registration of the symbols) is the same. We briefly mentioned this issue in the page 9 in the revised manuscript.

Comment #2: Thank you for updating the labels of the plots.

Response: We thank the Reviewer's correction.

Comment #3: Thank you for the experiment to help clarify the influence of SWNTs to your current-voltage characteristics.

Response: We thank the Reviewer's comment.

Comment #4: Thank you for the additional experiment to demonstrate lower voltage operation. Shouldn't this device then be the main feature of your manuscript? I understand you've already collected a lot of data with the higher voltage operation device. Is it difficult to obtain a suitable device yield with a 100 nm thick SiO_2 layer?

Response: We appreciate the Reviewer's comment. As the Reviewer mentioned, we have collected the most of the results based on OLEBs with 300-nm-thick SiO_2 insulators. We are currently working on demonstrating conductance visualization at the low operation voltage as

a follow-up study. We included a new fingerprint image obtained on an OLEB with a 100-nm-thick SiO₂ insulator in the inset of Supplementary Figure S10. We mentioned this issue in the page 12 in the revised manuscript.